# Optimization of infectious bronchitis virus-like particle expression in *Nicotiana benthamiana* as potential poultry vaccines

**Kamogelo M. Sepotokele**[1]*, **Martha M. O'Kennedy**[1,2], **Daniel B. R. Wandrag**[1], **Celia Abolnik**[1]

**1** Department of Production Animal Studies, Faculty of Veterinary Science, University of Pretoria, Pretoria, South Africa, **2** Biosciences, Council for Scientific and Industrial Research, Pretoria, South Africa

* sepotokelekamogelo@yahoo.com

**Data Availability Statement:** All relevant data are within the paper and its Supporting Information files.

## Abstract

Infectious bronchitis (IB) is a highly contagious, acute respiratory disease in chickens, with a severe economic impact on poultry production globally. The rapid emergence of regional variants of this *Gammacoronavirus* warrants new vaccine approaches that are more humane and rapid to produce than the current embryonated chicken egg-based method used for IB variant vaccine propagation (chemically-inactivated whole viruses). The production of virus-like particles (VLPs) expressing the Spike (S) glycoprotein, the major antigen which induces neutralizing antibodies, has not been achieved *in planta* up until now. In this study, using the *Agrobacterium*-mediated *Nicotiana benthamiana* (tobacco plant) transient expression system, the highest levels of VLPs displaying a modified S protein of a QX-like IB variant were obtained when the native transmembrane (TM) domain and cytoplasmic tail were substituted with that of the Newcastle disease virus (NDV) fusion glycoprotein, co-infiltrated with the NDV Matrix protein. In comparison, the native IB modified S co-infiltrated with IB virus membrane, envelope and nucleocapsid proteins, or substituted with the TM and CT of an H6-subtype influenza A virus hemagglutinin glycoprotein yielded lower VLP expression levels. Strong immunogenicity was confirmed in specific pathogen free chickens immunized intramuscularly with VLPs adjuvanted with Emulsigen®-P, where birds that received doses of 5 μg or 20 μg (S protein content) seroconverted after two weeks with mean hemagglutination inhibition titres of 9.1 and 10 $\log_2$, respectively. Plant-produced IB VLP variant vaccines are safer, more rapid and cost effective to produce than VLPs produced in insect cell expression systems or the traditional egg-produced inactivated whole virus oil emulsion vaccines currently in use, with great potential for improved IB disease control in future.

## Introduction

Infectious bronchitis (IB) is a highly contagious, acute respiratory disease in chickens that is listed by the World Organization for Animal Health (WOAH) due to its global economic impact, estimated to be only second to that of highly pathogenic avian influenza [1, 2]. The IB

**Funding:** This work was supported by the Department of Science and Innovation/ National Research Foundation (DSI/NRF) [SARChI grant numbers 114612, 129242]; and KMS received bursaries from the University of Pretoria, the DSI/ NRF and the HWSETA. The funders had no role in study design, data collection and analysis, decision to publish, or preparation of the manuscript.

**Competing interests:** The authors have declared that no competing interests exist.

virus (IBV) is a member of the *Coronaviridae* family, and the spherical or pleiomorphic virions, that range in size from 80 to 120 nm in diameter, have club-shaped glycoprotein spikes (S) of 16 to 21 nm projecting from the envelope surface [3–5]. The S glycoprotein is composed of two domains, with S1 (the bulb) facilitating attachment of the virus to the host cell receptor and displaying most of the viral antigenic epitopes that induce neutralising antibodies in the host, and S2 (the stalk) assisting with host cell attachment [5, 6].

Vaccination is commonly used to control IB in poultry, but a high virus mutation rate driven by genetic drift and genomic recombination has given rise to at least 32 global IBV genotypes [3, 7]. Novel variants are constantly emerging with the ability to escape the protection from commercial live attenuated and live recombinant IB vaccines, particularly if these mutations occur in the S1 gene's hypervariable (HVR) regions [8]. Inactivated whole virus oil emulsion vaccines derived from dominant circulating field strains remain one of the most effective control strategies for variant IBVs [9, 10] as they induce high titres of serum antibody that protects the kidneys, internal tissues and reproductive tract [4, 8]. Their production is however reliant on the isolation of field virus, a seed strain that is free of extraneous agents, adaptation to high growth properties in specific pathogen-free (SPF) chicken eggs by serial passage, and total chemical inactivation, all procedures requiring time (up to a year) and high containment conditions [10, 11].

Virus-like particles (VLPs) are multi-protein nanostructures that imitate native virus particles, but contain no genetic material, rendering them unable to replicate and cause infection, recombine with live viruses or revert to virulence [12, 13]. IBV VLPs can be produced more rapidly than other vaccine types and induce both cellular and humoral immunity [14–16]. Various IBV VLP permutations expressing native or chimeric S proteins were successfully expressed in insect cells using the baculovirus- mediated insect expression system, with immunogenicity or efficacy against live virus challenge demonstrated in the target animals [2, 17–19]. Plant-based platforms in turn offer numerous advantages over insect cells and other protein expression systems in terms of safety, speed, cost, and scalability, and VLPs for several viral families have been produced using the *Agrobacterium*-mediated *Nicotiana benthamiana* (tobacco plant) transient expression system [20–25]. No contaminating animal endotoxins or pathogens are produced, and the growth of infectious prions or viruses that infect humans is not supported [26]. The production of plant-produced VLPs can be achieved within two weeks of obtaining the antigenic gene sequence of interest, and scalability depends on the number of tobacco plants infiltrated. Mass agroinfiltration is achieved by vacuum infiltration, and larger volumes of plant material can be purified using methods such as ion exchange, size exclusion, or affinity chromatography, or tangential flow filtration [15, 25, 27].

Most recently, VLPs expressing the full-length S protein of the severe acute respiratory syndrome coronavirus 2 (SARS-CoV-2), a *Betacoronavirus*, were expressed in tobacco plants. VLP formation was achieved by introducing two consecutive proline residues into the loop between the S protein HVR-1 region and the central helix to stabilise the S protein's prefusion state [28], replacing the native signal sequence with a plant signal sequence, and the transmembrane (TM) domain and cytosolic tail (CT) with the equivalent sequences of an influenza A virus (IAV) H5-subtype hemagglutinin glycoprotein [29]. Although it was also reported that SARS-CoV-2 VLPs bearing the unmodified, native form of S alone or in combination with the envelope (E) and membrane (M) proteins express in *N. benthamiana* [30], this has not been the case with the *Gammacoronavirus* IB VLPs [31]. In the present study, we investigated incorporating stabilizing mutations in S and substituting the TM domain and CT from IAV or Newcastle disease virus (NDV), and successfully produced IB VLPs expressing the full-length S protein for the first time, that were shown to be highly immunogenic in the target species.

## Methods

### Recombinant plasmid design and construction

The S protein gene sequence for QX-like IBV strain ck/ZA/3665/11 (Genbank protein ID AKC34133; Uniprot ID A0A0E3XJ26) was modified by replacing the native signal peptide with a murine signal peptide to enhance expression [32], including a KOZAK sequence (GCCACC) upstream of the S1 domain, and removing the original endoplasmic reticulum (ER) retention signal (S1 Fig). Two consecutive proline residues were introduced at residues 843–844 into the loop between the first heptad repeat (HR1) and the central helix [28]. This construct, retaining the native IBV TM and CT domains, was designated mIBV-S2P (Fig 1 (C)). Ultramer primers were designed to substitute the TM domain and CT with that of the IAV H6 subtype hemagglutinin (HA) or the NDV F glycoprotein (LaSota strain). The constructs designated mIBV-S2P-IAV-H6$^{TM/CT}$, and mIBV-S2P-NDV-F$^{TM/CT}$ (Fig 1(C)) were created with the primer pairs in Table 1.

Gene sequences encoding the IBV M (Genbank protein ID AKC34136; Uniprot ID A0A0E3TI94), E (Genbank protein ID AKC34135; Uniprot ID A0A0E3TJA4) and N (Genbank protein ID AKC34140) proteins for strain ck/ZA/3665/11 were also chemically synthesised. A gene sequence encoding the Matrix protein of NDV isolate turkey/South Africa/N2057/2013, (Genbank protein ID KR815908) was already available. A plant codon-optimised gene sequence encoding the IAV M2 protein of strain A/New Caledonia/20/1999(H1N1) (Genbank protein ID HQ008884), was also already available for use. Based on our previous research [23, 33] we have found that chicken or human codon optimisation (GC-rich) enhances expression in plants, therefore all genes were chicken codon-optimised and GC-enriched unless otherwise stated and synthesised by BioBasic Inc., Canada, to incorporate *Age*I and *Xho*I sites on the 5' and 3' ends of each gene. The modified genes for mIBV--S2P-IAV-H6$^{TM/CT}$ and mIBV-S2P-NDV-F$^{TM/CT}$ were amplified by PCR. A mix containing 0.3 μM of each forward and reverse primer, dNTPs, 5 X HIFI buffer and HIFI DNA polymerase enzyme (KAPA Biosystems) with template DNA was prepared. PCR amplification was performed in the GeneAmp 2720 Thermocycler (Applied Biosystems) for 1 cycle of 95˚C initial denaturation for 3 min; 35 cycles of 98˚C denaturation for 20 sec, 70˚C annealing for 30 sec, 72˚C extension for 3.5 min and a final single 72˚C extension step for 5 minutes. The PCR products were separated on a 1% (m/v) agarose gel with a molecular weight marker. The target bands were excised and purified using the Zymoclean$^{TM}$ Gel DNA Recovery Kit. The purified genes for all constructs were digested using the *Age*I/*Xho*I restriction sites and ligated individually into pEAQ-HT plasmids (containing the silencing suppressor P19 [34]) with a Fast-link$^{TM}$ DNA Ligation Kit (Epicentre) according to the manufacturer's recommended procedure. Clones were verified by Sanger DNA sequencing at Inqaba Biotech (Pty), Ltd, Pretoria.

### Agroinfiltration

Recombinant plasmids for mIBV-S2P, mIBV-S2P-IAV-H6$^{TM/CT}$, and mIBV-S2P-NDV-F$^{TM/CT}$ were transformed into electrocompetent *Agrobacterium tumefaciens* strain AGL-1, ATCC® BAA-101$^{TM}$ and verified by colony PCR [23] and Sanger DNA sequencing. Glycerol stocks prepared from single colonies were propagated in Lysogeny Broth [35] (5 g/L yeast extract, 10 g/L tryptone, 10 g/L NaCl) with 50 μg/ml kanamycin, rifampicin (30 μg/ml) and carbenicillin (100 μg/ml) at 28˚C overnight (15–20 hours) until the OD$_{600}$ was ≥ 2. The same procedures were followed for the IBV M, E, and N recombinant plasmids, as well as for the NDV Matrix, and IAV M2 recombinant plasmids. Overnight cultures were centrifuged at 7000 x g for 7 minutes at 10˚C and the cell pellets were resuspended and diluted in 2-(N-morpholino)

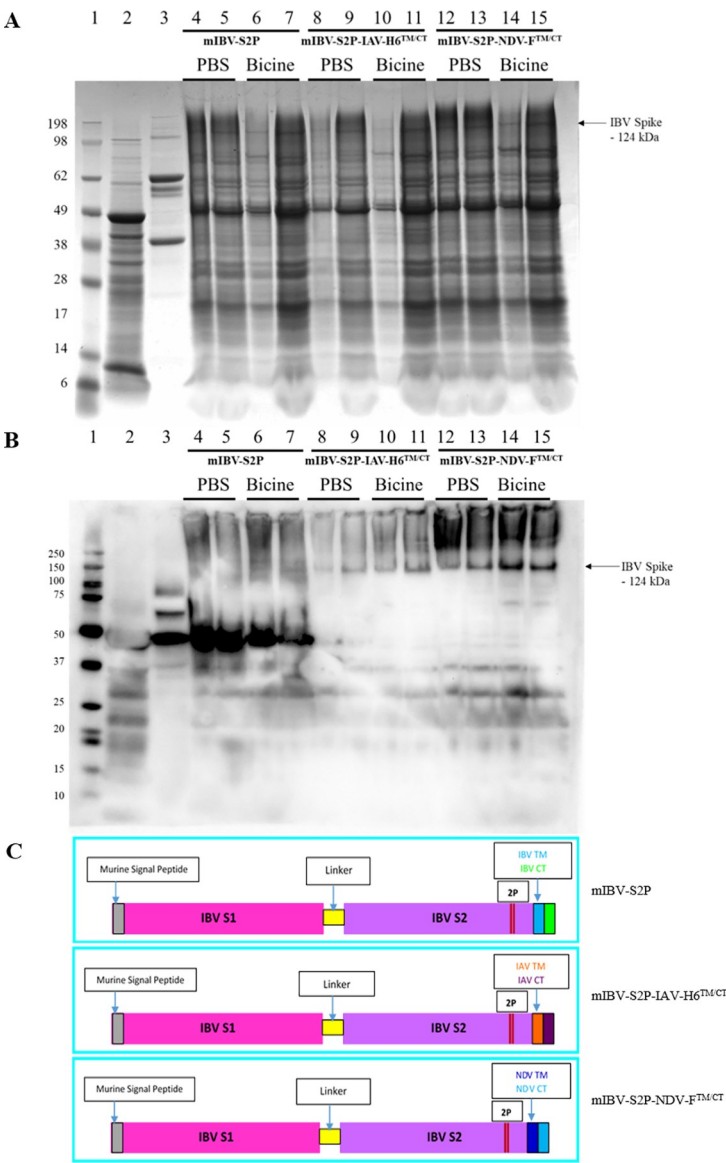

**Fig 1.** SDS-PAGE (**A**) and Western blot (**B**) of partially-purified plant-produced IBV S protein (Primary antibody–IBV antisera, secondary antibody—Goat-α-Chicken IgY HRP). Lane 1: molecular weight marker; Lane 2: plant-expressed empty pEAQ-HT vector; Lane 3: purified live QX-like IBV strain ck/ZA/3665/11; Lanes 4–7: mIBV-S2P:M: E:N fractions 2 (lanes 4 and 6) and 3 (lanes 5 and 7) extracted in either PBS or bicine as indicated; Lanes 8–11: mIBV-S2P-IAV-H6[TM/CT]:M2 fractions 2 (lanes 8 and 10) and 3 (lanes 9 and 11) extracted in either PBS or bicine as indicated; Lanes 12–15: mIBV-S2P-NDV-F[TM/CT]:NDV Matrix fractions 2 (lanes 12 and 14) and 3 (lanes 13 and 15) extracted in either PBS or bicine as indicated. (**C**) Schematic diagrams of recombinant Spike protein constructs mIBV-S2P with native IBV TM and CT, mIBV-S2P-IAV-H6[TM/CT] with TM and CT replaced with equivalent sequences of IAV H6 protein, and mIBV-S2P-NDV-F[TM/CT] with TM and CT replaced with equivalent sequences of NDV F protein.

ethanesulfonic acid (MES) infiltration buffer (10 mM MES, 10 mM MgCl$_2$, pH 5.6, 200 μM 3,5-Dimethoxy-4-hydroxy-acetophenone) to an OD$_{600}$ of between 1 and 2 before being combined to ratios of 2:1:1:1 (pEAQ-HT-mIBV-S2P: M: E: N), 2:1 (pEAQ-HT-mIBV--S2P-IAV-H6[TM/CT]: pEAQ-HT-IAV-M2) and 2:1 (pEAQ-HT-mIBV-S2P-NDV-F[TM/CT]: pEAQ-HT-NDV-Matrix). A higher ratio (4:1) of construct in relation to chaperone proteins

**Table 1.  Primers used for synthetic construct design.**

| Construct | Name | Sequence (5′– 3′) |
|---|---|---|
| mIBV-S2P-IAV-H6<sup>TM/CT</sup> | rIBV-S-H6<sup>TM/CT</sup> Forward | TTTACCGGTATGGGCTGGAGCTGG |
| | rIBV-S-H6<sup>TM/CT</sup> Reverse | AAACTCGAGTCAGATGCACACTCTGCACTGCATGCTGCCGTTGCTGCACATCCACAGGCCCATAG CAATGATCAGTCCCACCAGCACCAGGCTGCTGCTCACTGTGCTGGTAGATAGCCAGCTTAATGTAG GTTTTCAG |
| mIBV-S2P-NDV-F<sup>TM/CT</sup> | rIBV-S-H6<sup>TM/CT</sup> Forward | TTTACCGGTATGGGCTGGAGCTGG |
| | rIBV-S-F<sup>TM/CT</sup> Reverse | AAACTCGAGTCACATCTTTGTTGTGGCTCTCATCTGATCCAGGGTGTTATTGCCCAGCCACAGCAGT GTTTTCTGCTGTGCCTTCTGCTTGTACATCAGGTAGCAGGCCAGGATCAGGCTCAGGATTCCGAAC ACCAGGGAGATGATTGTCAGCACGATGTAGGTGATCAGCTTAATGTAGGTTTTCAG |

has also been shown to improve protein expression [36]. The solutions were left at room temperature (21°C) for 1–2 hours before infiltrating. *Agrobacterium*-transformed vector combinations were syringe-infiltrated into the leaves of 3–4-week-old *N. benthamiana* ΔXT/FT, a glycosylation mutant with a targeted downregulation of plant complex *N*-glycans β1,2-xylose and core α1,3-fucose that facilitate mammalian-like glycosylation [37]. The seeds had been germinated on Murashige and Skoog media and were hardened-off in pots containing sterilised compost/red soil/vermiculite (1:1:1) and fertilised with Chemicult hydroponic powder (Chemicult, South Africa) as per the manufacturer's recommendation. Plantlets were grown in a growth room facility fitted with alternating high sodium pressure and halide lamps, maintained at 26–28°C, for 16 h light and 8 h dark conditions for 3–4 weeks. The infiltrated leaves were harvested at 6 days post-infiltration (6 dpi has been reported for the harvest of plant-produced VLPs [20, 34], and was also previously determined to be the optimal harvest time for IB VLPs [31]) and weighed.

## Protein extraction and purification

The leaf tissue was extracted in 1 X PBS (140 mM NaCl, 1.5 mM KH$_2$PO$_4$, 10 mM Na$_2$HPO$_4$, 2.7 mM KCl, pH 7.4) or bicine buffers (50 mM Bicine, 20 mM NaCl pH 8.4) at 4°C supplemented with protease inhibitor cocktail (Sigma P2714) using a juicer as described by [23]. The mIBV-S2P, mIBV-S2P-IAV-H6<sup>TM/CT</sup>, and mIBV-S2P-NDV-F<sup>TM/CT</sup> VLPs were purified using sucrose density gradient centrifugation. Twenty and 70% (m/v) sucrose solutions were prepared by dissolving sterile sucrose in sterile 1X PBS, before overlaying 3 ml of 20% sucrose onto 2 ml of 70% sucrose in 38.5 ml ultra-clear<sup>TM</sup> Beckman tubes. The clarified leaf supernatant (33.5 ml) was layered gently on top of the sucrose gradient and centrifuged at 32 000 x g at 10°C for 2 hours. 0.5 ml fractions (successively labelled from fraction 1 upwards) were collected from the bottom of the 70% sucrose gradient using a tube coupled to a peristaltic pump.

## Analysis by SDS PAGE, Western blot and LC-MS/MS

Sucrose density gradient fractions 2 and 3 were analysed on 12% SDS-PAGE (sodium dodecyl sulfate–polyacrylamide gel electrophoresis) and Western blots were performed. For immunoblotting, the SDS-PAGE gel was transferred onto a Polyvinylidene difluoride (PVDF) membrane at 1.3 A, 25 V, for 7 minutes using the Trans-blot® Turbo™ Transfer system (BioRad) according to manufacturer's directions. Following overnight blocking in a 3% [m/v] Bovine Serum Albumin (BSA) in 1 X PBS blocking solution containing 0.1% [m/v] TWEEN20, the membrane was incubated in antisera (at a dilution of 1:1000) in blocking solution. The antisera was collected from a commercial flock that had been vaccinated twice with a combination of live commercial Mass-type vaccines (University of Pretoria). The membrane was washed and incubated in the secondary antibody, Goat-α-Chicken IgY HRP (Abcam, Novex) (at a dilution

of 1:2000) in 1 X PBS-TWEEN before protein detection with Clarity$^{TM}$ Western ECL chemilluminescence substrate (BioRad) and visualisation with a ChemiDoc$^{TM}$ MP Imaging System (Bio-Rad). Protein bands of the expected size were excised from stained SDS-PAGE gels and trypsin digested [38] for Liquid Chromatography Mass Spectrometry (LC-MS/MS) based peptide sequencing at CSIR Biosciences. The obtained MS/MS spectra were compared with the Uniprot Swissprot protein database through the Paragon search engine (AB Sciex) on Protein pilot v5. Only the proteins with a threshold of $\geq$ 99.9% confidence level were reported.

## Transmission electron microscopy

Sucrose density gradient partially purified IB VLPs fraction 3 (which displayed the most prominent bands on both SDS-PAGE and Western blot) for each of the plasmid combinations were adsorbed onto carbon-coated holey copper grids and stained as described in [23]. The grids were air-dried and imaged for VLPs using a JEOL JEM-1400 Flash Transmission Electron Microscope (TEM) at the University of Pretoria. One millilitre of SPF chicken egg allantoic fluid containing live strain ck/ZA/3665/11 (see Serological Testing) was clarified by centrifugation and also submitted for TEM.

## S protein quantitation

The fractions for the plasmid combination displaying the highest level of S protein expression in VLPs i.e., the mIBV-S2P-NDV-F$^{TM/CT}$: Matrix combination, as determined by the SDS-PAGE and immunoblotting results, were pooled and dialysed in PBS buffer using a 3500 mW CO Slide-A-Lyzer® Dialysis Cassette (ThermoFisher Scientific), after which 15% (m/v) trehalose dehydrate (Sigma-Aldrich) was added. The partially purified VLPs were quantified by densitometry of stained SDS-PAGE using Bovine Serum Albumin (BSA) protein standards of known concentrations and analysed using the quantification software on the ChemiDoc$^{TM}$ MP Imaging System (Bio-Rad).

## Immunization of chickens

Six-week-old SPF White Leghorn Chickens (*Gallus gallus*) (n = 20) purchased from Avi-Farms (Pty) Ltd, Pretoria were identified individually with numbered wing tags. The birds were housed in an isolation room in the Biosafety Level 3 facility at the University of Pretoria's Veterinary Faculty. Water and feed (Nova Feeds, South Africa) were provided *ad libitum*.

Two vaccine doses, 5 μg and 20 μg (S protein content) in single dose volumes of 0.25 ml were prepared for comparison. Just prior to immunization, the partially purified mIBV-S2P-NDV-F$^{TM/CT}$+NDV Matrix VLPs were diluted with sterile 1X PBS and vigorously mixed by shaking with 10% (v/v) of Emulsigen®-P Adjuvant (MVP, Phibro Animal Health, USA). The chickens were randomly assigned into two treatment groups of ten birds each. One millilitre blood samples collected from the wing veins of each of the birds prior to vaccination were used for the baseline comparison. The first group was immunized intramuscularly with the 5 μg VLP vaccine while the second group was immunized intramuscularly with the 20 μg VLP vaccine. After 14 days, blood samples were taken from all the chickens before euthanasia by cervical dislocation. All study procedures were approved by the Research and Animal Ethics Committees of the University of Pretoria (REC106-20), the CSIR (251/2018) and the Department of Agriculture, Land Reform, and Rural Development (12/11/1/1/MG).

## Serological testing

Sera were tested using two commercial IBV antibody detection kits, namely the BioChek Infectious Bronchitis Virus Antibody test kit (BioChek UK Ltd), and the IDEXX IBV Antibody test kit (IDEXX Laboratories Inc, United States) according to the manufacturers' recommendations. To prepare antigen for hemagglutination inhibition (HI) tests, SPF chicken egg allantoic fluid containing $10^{6.5}$ egg infectious dose $EID_{50}$/ 0.1 ml of live QX-like IBV strain chicken/ZA/3665/11 was centrifuged at 3000 rpm for 15 minutes at 4˚C and the clarified fluid was transferred to a clean tube. Five millilitres of the clarified fluid were mixed with 1 ml of a 1 U/ml neuraminidase solution from *Clostridium perfringens* (Abnova, Taiwan). Aliquots (1 millilitre) were incubated for overnight at 37˚C in a heating block, pooled and then chilled at 4˚C [39]. The HA and HI tests were performed according to the standard method using 1% (v/v) chicken red blood cells (CRBCs) [10]. The last well in the 2-fold titration of test serum where CRBC streaming in the tilted plate was observed, relative to the controls, was recorded as the $log_2$ HI titre. Due to limited antigen availability, HI was performed only on the blood samples taken two weeks after immunisation, PBS and SPF negative serum was used as the negative controls. HI results for the two treatment groups were tested for normality using the D'Agostino & Pearson test, Anderson-Darling test, Shapiro-Wilk test, and Kolmogorov-Smirnov test, and statistically compared using the non-parametric Mann-Whitney U Test in the GraphPad Prism v 9.4.1 software for Windows (La Jolla, CA, USA). A P value < 0.05 was considered significant.

# Results

## Agroinfiltration and protein expression

SDS PAGE of fractions 2 and 3 of the partially purified plant extracts showed bands at approximately 127 kDa for mIBV-S2P, 122 kDa for mIBV-S2P-IAV-H6$^{TM/CT}$, and 124 kDa for mIBV-S2P-NDV-F$^{TM/CT}$ correlating to the expected size of the IBV S protein, for all three constructs tested (Fig 1(A)) and LC-MS/MS based peptide sequencing confirmed the bands to be the S protein of IBV (S2 Fig). The S-protein specific bands detected by Western blot were the strongest for mIBV-S2P-NDV-F$^{TM/CT}$, followed by mIBV-S2P-IAV-H6$^{TM/CT}$, with the weakest expression of the S protein displayed by mIBV-S2P (Fig 1(B)). The large diffused bands >124 kDa are indicative of supramolecular structures associated with the VLPs as these are not present in negative controls, and similar results were reported by [40]. The chicken anti-IBV serum did not detect the S protein band in the QX-virus positive control, which was most likely because the quantity of S protein in the purified virus was low, but also because the source flock had been immunized with Mass-type IB vaccines, and there is little cross-protection between the highly variable S protein of different serotypes [41]. The IBV antisera however reacted strongly with the conserved 45 kDa IBV N protein in the purified positive control virus and the mIBV-S2P VLPs produced by co-infiltrating with the IBV M, E and N proteins, as N is a structural protein that is highly conserved between serotypes [42]. No difference was observed when bicine buffer was used compared to PBS to purify the VLPs. Under the TEM, VLPs were observed in the plant leaf extracts that resembled native IBV particles, ranging in diameter from 67 nm to 135 nm, with most being between 80 and 100 nm in size, and spikes ranging from 12 to 25 nm in length (Fig 2(A) and 2(B)). The crown-like S protein surface projections are clearly visible in both TEM images.

Since Western blot analysis showed that the mIBV-S2P-NDV-F$^{TM/CT}$ construct produced the highest levels of VLP formation and S protein expression, the VLPs formed from this construct were selected for the immunogenicity study. Fractions 2 and 3 were pooled, dialysed in

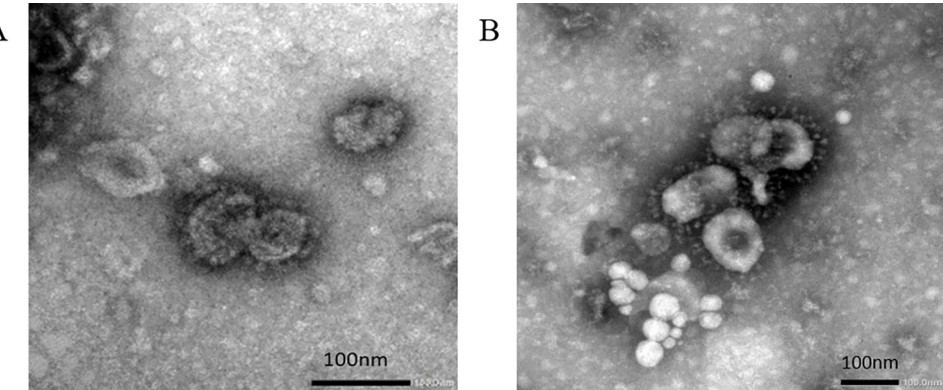

**Fig 2. Negative-stained transmission electron microscopy images of (A) live QX-like IBV strain ck/ZA/3665/11 and (B) IBV virus-like particles expressed with the mIBV-S2P-NDV-F$^{TM/CT}$ construct in *N. benthamiana*.** The crown-like S protein surface projections are clearly visible in both TEM images.

PBS and stabilised with trehalose. SDS PAGE was performed, and densitometry analysis determined the concentration of the 124 kDa S protein band to be ~ 101 ng/ μl (S3 Fig). As with live strain ck/ZA/3665/11 on which the VLP designs were based and most strains of IBV, the tobacco plant-produced IB VLPs did not agglutinate CRBCs, therefore HA could not be used for the vaccine quantitation. Following treatment with a neuraminidase solution [39], haemagglutination ability of live ck/ZA/3665/11 virus could be activated, allowing for the haemagglutination inhibition study to be performed.

## Immunogenicity in chickens

All 20 pre-vaccination serum samples tested negative for the presence of IBV-specific antibodies with a commercial ELISA kit, verifying that the SPF chickens had no prior exposure to IBV. Two weeks after vaccination, the ten chickens vaccinated with 5 μg seroconverted with HI titres that ranged from 8 $\log_2$ to 10 $\log_2$ with a geometric mean titre (GMT) of 9.1 $\log_2$ whereas the ten that received 20 μg had significantly higher (P = 0.0001) HI titres that ranged from 10 $\log_2$ to 12 $\log_2$ with a GMT of 10.5 $\log_2$ (Fig 3). All twenty samples collected at 14 days tested negative for the presence of IBV-specific antibodies on two commercial ELISA kits, and no adverse vaccination effects at the injection site were observed in any of the chickens during the course of the experiment.

## Discussion

In this study, VLPs displaying the immunogenic S protein of IB virus were successfully produced *in planta* for the first time. Previously, the native IBV S protein with various combinations of modifications including replacement of the native retention peptide with a murine signal peptide, substitution of the TM and/or CT domains with those of IAV HA, as well as removal of the ER retention signal, did not successfully assemble VLPs in plants [31]. The stabilising proline residues in the S protein introduced in the present study therefore appears to be a critical requirement for plant-produced IB VLPs, and the highest levels of IB VLP expression (as confirmed by Western blot and LC-MS/MS) were obtained when the native TM domain and CT of the S glycoprotein were substituted with the equivalent sequence of NDV F glycoprotein, co-infiltrated with the NDV Matrix protein. Slightly lower levels of VLP expression were obtained when the chimeric modified S protein contained the TM and CT of an

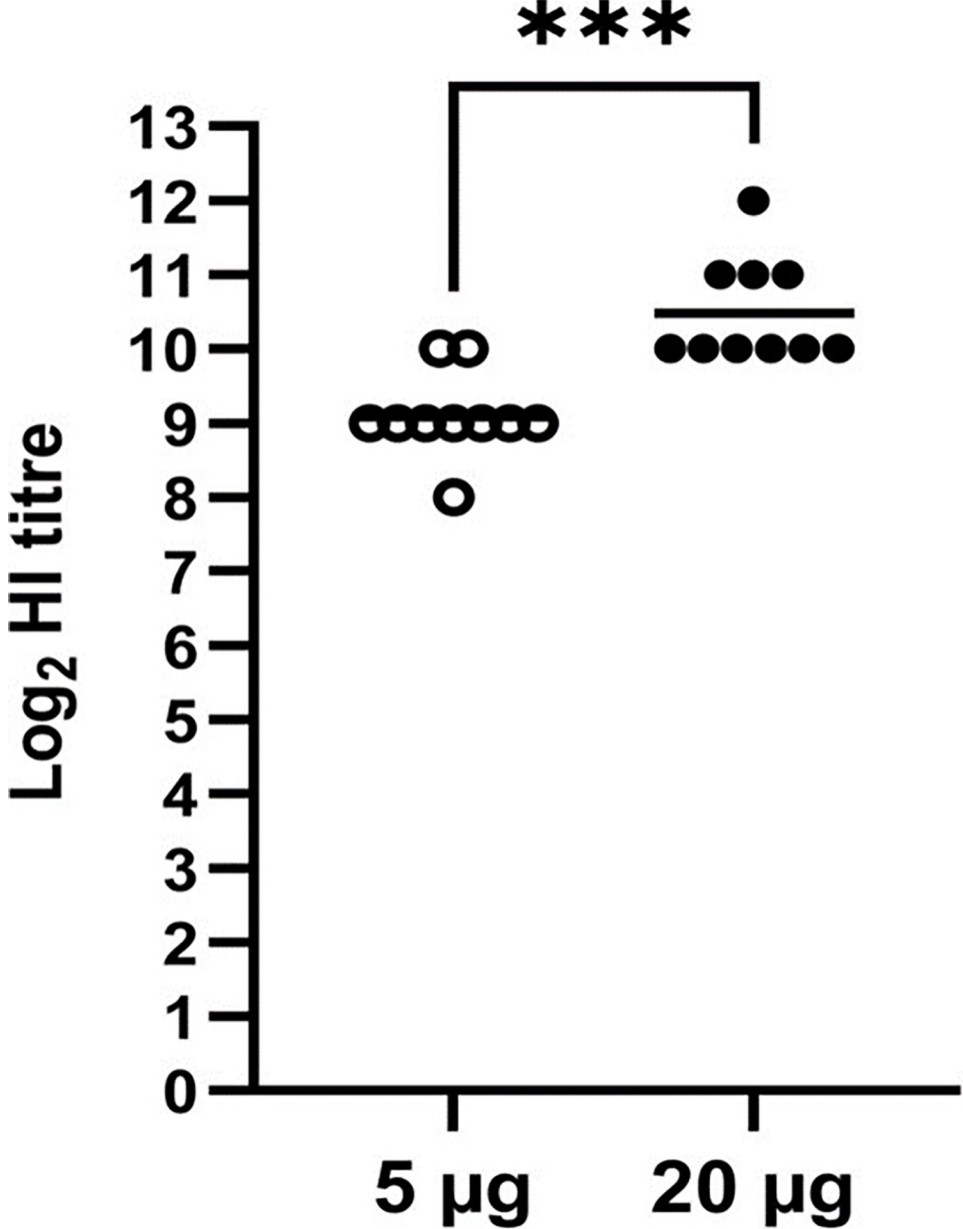

**Fig 3. Hemagglutination inhibition (HI) titres for chicken sera after a single vaccination with 5 and 20 μg doses of IB VLPs.** The bar depicts the geometric mean titre; statistical significance is indicated by the asterisks ($P < 0.05$). A titre of 4 log$_2$ (1:16) or higher is considered positive for the HI test.

H6-subtype IAV hemagglutinin protein, as confirmed by Western blot and LC-MS/MS. The lowest levels of VLP expression were obtained when the modified S protein with its native TM and CT was co-infiltrated with the IBV M, E and N structural proteins, but in general co-infiltrating with virus-specific structural proteins enhanced VLP formation.

The yield of the optimally-expressed VLP (developed from the mIBV-S2P-NSV-F$^{TM/CT}$ construct) in this study was calculated at 16.77 mg/kg of leaf material harvested. The NDV Matrix protein plays a key role in the assembly and budding of viruses at the cell membrane [43], suggesting that its presence may improve expression of the IBV S protein, perhaps even

more so when co-expressing with the S protein at a higher ratio (4:1), as suggested by [36]. It was previously shown that the transient co-expression of IAV M2 has the potential to improve the accumulation and the stability of recombinant proteins in the secretory pathway of plant cells through pH regulation [36], but co-infiltrating with the IAV M1 protein, was not necessary for and may decrease the formation of IAV VLPs in plants [44].

The IB VLPs with the chimeric S-NDV protein + NDV Matrix protein, that displayed the most abundant VLP expression under TEM and the strongest antigenicity *in vitro*, was formulated as a vaccine with Emulsigen®-P adjuvant. Emulsigen®-P stimulates optimal immune responses in chickens vaccinated with plant-produced IAV VLPs and is used at a lower volume compared to other adjuvants [33]. The SPF chickens immunized with 5 µg (approximate S protein content) doses of the adjuvanted IB VLPs showed no adverse vaccine effects, whereas the traditional whole inactivated virus oil emulsion vaccines occasionally cause lesions at the injection site that lead to carcass downgrades [45]. A single 5 µg dose of per bird of the plant-produced IB VLP vaccine was sufficient to elicit a strong humoral response in SPF chickens of S-specific antibodies using the HI test, but there is scope to improve the quantitation methods of non-agglutinating VLPs such as this to further optimise the effective dose.

Since the vaccine did not contain the IBV structural proteins (M, N, E), no antibodies were detected by two commercial IBV ELISA kits, which are targeted at detecting the highly conserved structural IBV group-specific antibodies common to all IBV serotypes. This feature of the plant-produced IB VLPs can potentially be used for DIVA if only VLP vaccines are administered, but in practice, the vaccination program would likely include a combination of heterologous live-attenuated and homologous VLP vaccines, that affords the best protection. The HI test on the other hand, was able to detect the neutralizing S-protein specific antibodies elicited by the VLP vaccine. The efficacy of the IBV VLP vaccine produced here, for which the immunogenicity and lack of toxicity in the target species was demonstrated, will be assessed by live virus challenge in a follow-up study.

## Supporting information

**S1 Fig. Protein sequence of the synthetic gene mIBV-S2P.** The murine signal peptide is highlighted in blue, the linker in magenta, and the S2 domain in grey with heptad repeat 1 in yellow, the central helix in green, and the two stabilizing proline substitutions in boldface and underlined.
(DOCX)

**S2 Fig.** Protein confirmation using LC-MS/MS-based peptide sequencing of the modified IBV spike protein constructs compared in this study (**A**) mIBV-S2P, (**B**) mIBV-S2P-IAV-H6$^{TM/CT}$, and (**C**) mIBV-S2P-NDV-F$^{TM/CT}$. The percentage sequence coverage is indicated above with several unique peptides identified with > 90% confidence. Peptides with > 95% confidence are highlighted in green, those with 50–95% confidence in yellow, and those with <50% confidence in red. No peptides were identified for the non-highlighted regions of the sequence (grey).
(DOCX)

**S3 Fig. Densitometric analysis by SDS-PAGE of partially-purified mIBV-S2P-NDV-F$^{TM/CT}$ VLPs.** Lane 1: SeeBluePlus2 protein ladder; Lane 2: BSA Standard 100 ng/µl; Lane 3: BSA Standard 150 ng/µl; Lane 4: BSA Standard 200 ng/µl; Lane 5: BSA Standard 250 ng/µl; Lane 6: BSA Standard 300 ng/µl; Lane 7: Dialysed VLP sample (25 µl); Lane 8: Dialysed VLP sample (10 µl); Lane 9: PageRuler Prestained protein ladder; Lane 10: Positive control (Live QX-like

IBV); Lane 11: Negative control (pEAQ-HT-empty).
(DOCX)

**S4 Fig. Original uncropped, unedited SDS-PAGE of partially-purified plant-produced IBV S protein Lane 1: molecular weight marker; Lane 2: plant-expressed empty pEAQ-HT vector; Lane 3: purified live QX-like IBV strain ck/ZA/3665/11; Lanes 4–7: mIBV-S2P:M:E:N fractions 2 (lanes 4 and 6) and 3 (lanes 5 and 7) extracted in either PBS or bicine as indicated; Lanes 8–11: mIBV-S2P-IAV-H6$^{TM/CT}$:M2 fractions 2 (lanes 8 and 10) and 3 (lanes 9 and 11) extracted in either PBS or bicine as indicated; Lanes 12–15: mIBV-S2P-NDV-F$^{TM/CT}$:NDV Matrix fractions 2 (lanes 12 and 14) and 3 (lanes 13 and 15) extracted in either PBS or bicine as indicated.**
(PNG)

**S5 Fig. Original uncropped, unedited Western blot (B) of partially-purified plant-produced IBV S protein (Primary antibody–IBV antisera, secondary antibody—Goat-α-Chicken IgY HRP).** Lane 1: molecular weight marker; Lane 2: plant-expressed empty pEAQ-HT vector; Lane 3: purified live QX-like IBV strain ck/ZA/3665/11; Lanes 4–7: mIBV-S2P:M:E:N fractions 2 (lanes 4 and 6) and 3 (lanes 5 and 7) extracted in either PBS or bicine as indicated; Lanes 8–11: mIBV-S2P-IAV-H6$^{TM/CT}$:M2 fractions 2 (lanes 8 and 10) and 3 (lanes 9 and 11) extracted in either PBS or bicine as indicated; Lanes 12–15: mIBV-S2P-NDV-F$^{TM/CT}$:NDV Matrix fractions 2 (lanes 12 and 14) and 3 (lanes 13 and 15) extracted in either PBS or bicine as indicated.
(TIF)

**S1 Raw images.**
(PDF)

## Acknowledgments

The authors thank Antoinette Lensink, Karen Ebersohn, Danielle Henn, Albert Mabetha, Sharon Kgasago, Thandeka Phiri, Thlasila Aphane, Alma Truyts, and Sipho Mamphutha for technical assistance. Tanja Smith provided the pEAQ-HT-NDV-Matrix and pEAQ-HT-H1N1-M2 constructs. The pEAQ-HT vector was used under a research license from Plant Biosciences Ltd, UK. The MVP Emulsigen-P adjuvant was donated by Phibro Animal Health (USA).

## Author Contributions

**Conceptualization:** Kamogelo M. Sepotokele, Martha M. O'Kennedy.

**Data curation:** Kamogelo M. Sepotokele, Celia Abolnik.

**Formal analysis:** Kamogelo M. Sepotokele, Celia Abolnik.

**Funding acquisition:** Celia Abolnik.

**Investigation:** Kamogelo M. Sepotokele, Martha M. O'Kennedy, Daniel B. R. Wandrag, Celia Abolnik.

**Methodology:** Kamogelo M. Sepotokele, Martha M. O'Kennedy, Celia Abolnik.

**Resources:** Martha M. O'Kennedy, Celia Abolnik.

**Supervision:** Martha M. O'Kennedy, Celia Abolnik.

**Writing – original draft:** Kamogelo M. Sepotokele, Celia Abolnik.

**Writing – review & editing:** Kamogelo M. Sepotokele, Martha M. O'Kennedy, Daniel B. R. Wandrag, Celia Abolnik.

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
