## [Decision Letter · Decision Letter 0]

24 Mar 2023

PONE-D-23-05238Optimization of Infectious Bronchitis virus-like particle expression in Nicotiana benthamiana as potential poultry vaccinesPLOS ONE

Dear Dr. Sepotokele

Thank you for submitting your manuscript to PLOS ONE. After careful consideration, we feel that it has merit but does not fully meet PLOS ONE’s publication criteria as it currently stands. Therefore, we invite you to submit a revised version of the manuscript that addresses the points raised during the review process.

We look forward to receiving your revised manuscript.

Kind regards,

Haitham Mohamed Amer, PhD

Academic Editor

PLOS ONE

Journal Requirements:

In your cover letter, please note whether your blot/gel image data are in Supporting Information or posted at a public data repository, provide the repository URL if relevant, and provide specific details as to which raw blot/gel images, if any, are not available. Email us at plosone@plos.org if you have any questions

Reviewers' comments:

Reviewer's Responses to Questions

**Comments to the Author**

1. Is the manuscript technically sound, and do the data support the conclusions?

Reviewer #1: Partly

Reviewer #2: Partly

2. Has the statistical analysis been performed appropriately and rigorously? 

Reviewer #1: I Don't Know

Reviewer #2: I Don't Know

3. Have the authors made all data underlying the findings in their manuscript fully available?

Reviewer #1: No

Reviewer #2: Yes

4. Is the manuscript presented in an intelligible fashion and written in standard English?

Reviewer #1: Yes

Reviewer #2: Yes

5. Review Comments to the Author

Reviewer #1: This paper describes the production of VLPs of IBV in plants using transient expression. Various different gene constructs are tested and the highest yielding one is used for an immunogenicity study in target animals. On the whole, this manuscript requires some modifications to improve clarity (especially with regards to the different constructs used and what Figure 1 shows). Finally the immunogenicity experiment described in figure 3 is lacking in any controls. More detailed points are raised below:

- The authors should add a figure that includes labelled diagrams for the constructs expressed in this paper.

- Figures S1 and S2 seems to have been swapped.

- Both supplementary figures appear to be TIFF image files, but they show sequence information. These files should therefore be in Word or PDF format so that readers can perform the “copy” function for BLAST searches, sequence analysis, etc.

- Figure 1: What is “fractions 2 and 3” referred to multiple times in the text and figure legend? Is this fractions from the sucrose cushion? If so what part of the gradient? The authors need to explain this clearly or show a diagram. Also the figure legend is unnecessarily confusing. If lanes 4-7 are from the same construct, then write something along the lines of “Lanes 4-7: mIBV-S2P:M:E:N fractions 2 (lanes 4 and 6) and 3 (lanes 5 and 7) extracted in either PBS or Bicine, as indicated”. Or add a line of labels in the figure itself above the wells for “fraction”, and another line for “construct”. In fact I would strongly recommend adding a “construct” line of labels to the images in Figure 1 regardless of changes to the figure legend. Same thing for lanes 8-11 and 12-15.

- Line 288-289: Gel densitometry is a rather poor method of protein quantification, a superior alternative would have been a modified Lowry assay or BCA assay. Because the authors chose to use gel densitometry, they need to show an image of the SDS-PAGE gel that was used for quantitation.

- Line 290: Is this expected? Does live IBV cause agglutination of CRBCs (I assume the answer is yes given how the subsequent immunogenicity assay was performed)? Do currently used IBV vaccines cause agglutination of CRBCs? The authors should indicate this for context and discuss.

- Line 299: All twenty samples tested negative for what?

- Figure 3 and generally vaccination experiment: there are no controls in this experiment. This experiment should have a positive control (chickens vaccinated with a currently licensed vaccine) and a negative control (PBS + adjuvant, or, even better, chickens vaccinated with plant extract prepared and purified in the same way as the vaccine but without expression of the antigen + adjuvant). The authors should acknowledge this methodological weakness in the discussion and indicate that it makes any conclusion about immunogenicity preliminary rather than solid. I certainly hope that the live virus challenge that the authors mention will be performed with appropriate controls.

Reviewer #2: In this manuscript, the authors describe a method by using Nicotiana benthamiana expression system to express IBV S protein and other viral proteins to produce virus like particles (VLPs). By co-expressing IBV S, N, M, E, or IBV S-NDV F (TMCM), NDV M, or IBV S-AIV H6(TMCM), AIV M2, the authors found the co-expressing group of IBV S-NDV F (TMCM), NDV M obtains the highest level of VLPs. This is a good attempt by using plant to produce IB VLPs as vaccine candidate, with lower cost and safer than the baculovirus insect cell expression system. The idea is novel; however, there is some issues need to be addressed in this study.

1. The results and figures legend interpret are not clear. Please check each figure’s interpret and give more detailed and clear information. For example, which antibody was applied for Fig. 1B? How was the HI result obtained in Fig. 3?

2. It seems the antibody for IBV is not sensitive for S protein detection. In Fig. 1B, there is no clear band for S protein in lane 3 (positive control). Also, it seems the membrane transfer has problem, the middle part (lane 8, 9, 10, 11, 12) membrane transfer is not successful, result in the white signal. Fig 1A, the arrow indicate S protein band also appears in the negative control (lane 2). Thus, it is not convincing to claim that the band is S protein. Usually, the glycol-S protein size is much bigger than 124 kDa. Please re-do the Western blot by using more specific S protein antibody. Current data is not convincing to show the successful expression of S protein. Moreover, please quantify the S protein containing VLPs.

3. The is no S antibody measurement in the chicken serum after immunization. Please use ELISA kit to measure the kinetic induction of S protein antibody in the serum.

4. Please do the QX strain attacking experiment to show the VLP protection efficiency, by using the H120 vaccination as control.

---

## [Author Response · Author response to Decision Letter 0]

6 Apr 2023

Response to Reviewers

Journal Requirements:

RESPONSE: The formatting has been revised as required. 

In your cover letter, please note whether your blot/gel image data are in Supporting Information or posted at a public data repository, provide the repository URL if relevant, and provide specific details as to which raw blot/gel images, if any, are not available. Email us at plosone@plos.org if you have any questions

RESPONSE: The original images have been uploaded as a supporting file (S1_raw_images).

3. We note that you have included the phrase “data not shown” in your manuscript. Unfortunately, this does not meet our data sharing requirements. PLOS does not permit references to inaccessible data. We require that authors provide all relevant data within the paper, Supporting Information files, or in an acceptable, public repository. Please add a citation to support this phrase or upload the data that corresponds with these findings to a stable repository (such as Figshare or Dryad) and provide and URLs, DOIs, or accession numbers that may be used to access these data. Or, if the data are not a core part of the research being presented in your study, we ask that you remove the phrase that refers to these data

RESPONSE: All the relevant data is actually presented, so the phrase has been replaced with an appropriate citation (lines 150-153), (lines 337-341), or was removed where we referred to our previous unpublished laboratory findings (lines 345-346).

 

Reviewers' comments:

Reviewer's Responses to Questions

Comments to the Author

1. Is the manuscript technically sound, and do the data support the conclusions?

Reviewer #1: Partly

Reviewer #2: Partly

2. Has the statistical analysis been performed appropriately and rigorously?

Reviewer #1: I Don't Know

Reviewer #2: I Don't Know

3. Have the authors made all data underlying the findings in their manuscript fully available?

Reviewer #1: No

Reviewer #2: Yes

4. Is the manuscript presented in an intelligible fashion and written in standard English?

Reviewer #1: Yes

Reviewer #2: Yes

5. Review Comments to the Author

Reviewer #1: This paper describes the production of VLPs of IBV in plants using transient expression. Various different gene constructs are tested and the highest yielding one is used for an immunogenicity study in target animals. On the whole, this manuscript requires some modifications to improve clarity (especially with regards to the different constructs used and what Figure 1 shows). Finally the immunogenicity experiment described in figure 3 is lacking in any controls. More detailed points are raised below:

- The authors should add a figure that includes labelled diagrams for the constructs expressed in this paper.

RESPONSE: The requested diagrams depicting constructs has been added as supplementary file (S3_Fig).

- Figures S1 and S2 seems to have been swapped.

RESPONSE: Thank you for picking this up, the figure titles have been amended accordingly. 

- Both supplementary figures appear to be TIFF image files, but they show sequence information. These files should therefore be in Word or PDF format so that readers can perform the “copy” function for BLAST searches, sequence analysis, etc.

RESPONSE: Editable docx versions of these figures have been uploaded with the revised manuscript.

- Figure 1: What is “fractions 2 and 3” referred to multiple times in the text and figure legend? Is this fractions from the sucrose cushion? If so what part of the gradient? The authors need to explain this clearly or show a diagram. Also the figure legend is unnecessarily confusing. If lanes 4-7 are from the same construct, then write something along the lines of “Lanes 4-7: mIBV-S2P:M:E:N fractions 2 (lanes 4 and 6) and 3 (lanes 5 and 7) extracted in either PBS or Bicine, as indicated”. Or add a line of labels in the figure itself above the wells for “fraction”, and another line for “construct”. In fact I would strongly recommend adding a “construct” line of labels to the images in Figure 1 regardless of changes to the figure legend. Same thing for lanes 8-11 and 12-15.

RESPONSE: To clarify the concept of fractions, lines 176-178 were amended as follows: “0.5 ml fractions (successively labelled from fraction 1 upwards) were collected from the bottom of the 70 % sucrose gradient using a tube coupled to a peristaltic pump”. Figure legends were amended as suggested for clarity, and additional “construct” line was added above the wells in Fig1 as suggested. 

- Line 288-289: Gel densitometry is a rather poor method of protein quantification, a superior alternative would have been a modified Lowry assay or BCA assay. Because the authors chose to use gel densitometry, they need to show an image of the SDS-PAGE gel that was used for quantitation.

RESPONSE: Yes, we agree that Lowry and BCA are superior methods for protein quantitation, but these can only be used to determine the total protein content. The protein extract here was intentionally partially purified because highly-purified VLP would drive up the production cost of such a veterinary vaccine, and it was important to test a preparation that would ultimately be commercially viable. The S-protein is the protective antigen, so we specifically needed an approximation of its concentration to calculate the vaccine dose, and we were therefore restricted to using densitometry of a PAGE gel. The densitometry gel has been added as a supplementary figure (S4_Fig).

- Line 290: Is this expected? Does live IBV cause agglutination of CRBCs (I assume the answer is yes given how the subsequent immunogenicity assay was performed)? Do currently used IBV vaccines cause agglutination of CRBCs? The authors should indicate this for context and discuss.

RESPONSE: Some live IBVs can agglutinate CRBCs in their native form but many (possibly the majority) cannot. Yes, this was expected because the native live IBV strain ck/ZA/3665/11 used in the design was unable to agglutinate CRBCs without neuraminidase pre-treatment, therefore we did not expect that the VLP would agglutinate CRBCs either but we tested it anyway. Live IBV strain ck/ZA/3665/11 could be treated with neuraminidase to enable it to be used as an antigen in the HI tests, and yes, we did try the neuraminidase treatment on the VLP without success, but the neuraminidase may not work on plant-produced proteins as with avian-produced antigens, and the heat inactivation step of the neuraminidase may also have a detrimental effect on the VLP. 

Lines 296-302 were amended to better reflect the abovementioned, and with the reference of Ruano et al., 2000 included. 

- Line 299: All twenty samples tested negative for what?

RESPONSE: Line 310 was amended to reflect that all twenty samples tested negative for the presence of IBV-specific antibodies. 

- Figure 3 and generally vaccination experiment: there are no controls in this experiment. This experiment should have a positive control (chickens vaccinated with a currently licensed vaccine) and a negative control (PBS + adjuvant, or, even better, chickens vaccinated with plant extract prepared and purified in the same way as the vaccine but without expression of the antigen + adjuvant). The authors should acknowledge this methodological weakness in the discussion and indicate that it makes any conclusion about immunogenicity preliminary rather than solid. I certainly hope that the live virus challenge that the authors mention will be performed with appropriate controls.

RESPONSE: Thank you for the comment, but please bear in mind that there is no methodological weakness here because the clinical part of this study is not a vaccination experiment (which by definition entails a challenge with live virus), it is instead an immunogenicity experiment to verify that the VLPs do induce IBV-specific antibodies in the target species. Our Animal Ethics Committee was also insistent that this pilot be performed prior to the official vaccination study, not only to establish immunogenicity, but also toxicity. The pre-vaccination titres of the chickens served as the negative control here; there would be little point in including and ultimately sacrificing a group of non-immunized birds. We are compelled to reduce animal use. Yes, indeed we included appropriate control groups in the official vaccine study that followed this one. Lines 370-371 were modified to reiterate that this was an immunogenicity study that precedes the vaccine efficacy study.

 

Reviewer #2: In this manuscript, the authors describe a method by using Nicotiana benthamiana expression system to express IBV S protein and other viral proteins to produce virus like particles (VLPs). By co-expressing IBV S, N, M, E, or IBV S-NDV F (TMCM), NDV M, or IBV S-AIV H6(TMCM), AIV M2, the authors found the co-expressing group of IBV S-NDV F (TMCM), NDV M obtains the highest level of VLPs. This is a good attempt by using plant to produce IB VLPs as vaccine candidate, with lower cost and safer than the baculovirus insect cell expression system. The idea is novel; however, there is some issues need to be addressed in this study.

1. The results and figures legend interpret are not clear. Please check each figure’s interpret and give more detailed and clear information. For example, which antibody was applied for Fig. 1B? How was the HI result obtained in Fig. 3?

RESPONSE: Figure 1’s legend has been amended for clarity. Primary and secondary antibody now included in the figure legend, details of which can be found in the methodology, lines 182-192). Lines 242-247 describe the haemagglutination inhibition test (with reference to WOAH, 2019 for the standard method) used to obtain the HI results. 

2. It seems the antibody for IBV is not sensitive for S protein detection. In Fig. 1B, there is no clear band for S protein in lane 3 (positive control). Also, it seems the membrane transfer has problem, the middle part (lane 8, 9, 10, 11, 12) membrane transfer is not successful, result in the white signal. Fig 1A, the arrow indicate S protein band also appears in the negative control (lane 2). Thus, it is not convincing to claim that the band is S protein. Usually, the glycol-S protein size is much bigger than 124 kDa. Please re-do the Western blot by using more specific S protein antibody. Current data is not convincing to show the successful expression of S protein. Moreover, please quantify the S protein containing VLPs.

RESPONSE: Yes, the S protein was not detected in the positive control. The most likely reason for this was that the quantity of S protein in the purified virus used for the positive control was low. The flock that the protein was purified from was also immunized with Mass-type IBV vaccines, and the level of cross-protection is low between the spike protein from different serotypes (Cavanagh, 2003). This is already mentioned in lines 265-268.

While a similar sized- band appears in the negative control, this is most likely a plant protein of a similar size. Most importantly, the corresponding bands in the relevant lanes, were excised from the gel (in lanes 5, 9, and 13, correlating to each construct) were analysed by LC-MS/MS (peptide sequencing), which confirmed the presence of the IBV spike protein in high enough quantities to confidently confirm that the S protein is present in those bands. The SDS-PAGE, Western blot, and LC-MS/MS results taken together allow us to confidently make the conclusion that the S-protein was well-expressed in the plants, and this was also subsequently verified by the inducement of IBV S-specific antibodies in the immunogenicity study in chickens. The gel used for densitometry quantification of the VLP S-protein concentration has been included as a supplementary figure (S4_Fig).

3. The is no S antibody measurement in the chicken serum after immunization. Please use ELISA kit to measure the kinetic induction of S protein antibody in the serum.

RESPONSE: There is no validated commercial ELISA available that measures IBV S-antibodies, presumably because of the vast number of IBV serotypes present globally and the propensity of IBV for antigenic drift. Developing and validating an S-specific ELISA for the QX-like virus on which the VLP designs were based was far beyond the scope of this study. Therefore, only HI could be used to measure specific S-antibody responses.

4. Please do the QX strain attacking experiment to show the VLP protection efficiency, by using the H120 vaccination as control.

RESPONSE: We are uncertain about what the reviewer means by an “attacking” experiment, but thought that he/she is referring to virus neutralization tests (with the H120 vaccine). Please bear in mind that the clinical part of this study was designed and intended only as an immunogenicity experiment, to demonstrate that the VLPs were immunogenic and non-toxic in the target species. We verified both, and in the vaccine study that follows this one, we officially evaluated the efficacy of the VLPs and the ability of the vaccine to reduce virus shedding and protect against disease; no further changes made.

---

## [Decision Letter · Decision Letter 1]

23 May 2023

PONE-D-23-05238R1Optimization of Infectious Bronchitis virus-like particle expression in Nicotiana benthamiana as potential poultry vaccinesPLOS ONE

Dear Dr. Sepotokele,

Thank you for submitting your manuscript to PLOS ONE. After careful consideration, we feel that it has merit but does not fully meet PLOS ONE’s publication criteria as it currently stands. Therefore, we invite you to submit a revised version of the manuscript that addresses the points raised during the review process.

 Please submit your revised manuscript by Jul 07 2023 11:59PM. If you will need more time than this to complete your revisions, please reply to this message or contact the journal office at plosone@plos.org. Please include the following items when submitting your revised manuscript:A rebuttal letter that responds to each point raised by the academic editor and reviewer(s). You should upload this letter as a separate file labeled 'Response to Reviewers'.A marked-up copy of your manuscript that highlights changes made to the original version. You should upload this as a separate file labeled 'Revised Manuscript with Track Changes'.An unmarked version of your revised paper without tracked changes. You should upload this as a separate file labeled 'Manuscript'.

We look forward to receiving your revised manuscript.

Kind regards,

Haitham Mohamed Amer, PhD

Academic Editor

PLOS ONE

Reviewers' comments:

Reviewer's Responses to Questions

**Comments to the Author**

1. If the authors have adequately addressed your comments raised in a previous round of review and you feel that this manuscript is now acceptable for publication, you may indicate that here to bypass the “Comments to the Author” section, enter your conflict of interest statement in the “Confidential to Editor” section, and submit your "Accept" recommendation.

Reviewer #1: All comments have been addressed

Reviewer #2: (No Response)

2. Is the manuscript technically sound, and do the data support the conclusions?

Reviewer #1: Yes

Reviewer #2: No

3. Has the statistical analysis been performed appropriately and rigorously? 

Reviewer #1: I Don't Know

Reviewer #2: I Don't Know

4. Have the authors made all data underlying the findings in their manuscript fully available?

Reviewer #1: Yes

Reviewer #2: Yes

5. Is the manuscript presented in an intelligible fashion and written in standard English?

Reviewer #1: Yes

Reviewer #2: Yes

6. Review Comments to the Author

Reviewer #1: (No Response)

Reviewer #2: In this manuscript, the authors describe a method by using Nicotiana benthamiana expression system to express IBV S protein and other viral proteins to produce virus like particles (VLPs). By co-expressing IBV S, N, M, E, or IBV S-NDV F (TMCM), NDV M, or IBV S-AIV H6(TMCM), AIV M2, the authors found the co-expressing group of IBV S-NDV F (TMCM), NDV M obtains the highest level of VLPs. This is a good attempt by using plant to produce IB VLPs as vaccine candidate, with lower cost and safer than the baculovirus insect cell expression system. The idea is novel; however, there is some issues still need to be addressed in this study.

1. The author should re-perform the Western blot in Fig 1b and get better quality figure. The current Western blot figure quality is not acceptable, as there is no clear S protein band, even in the positive control (lane 3).

2. S antibody measurement in the chicken serum after immunization is necessary.

3. The QX strain infection experiment after immunization is necessary to show the VLP protection efficiency, by using the H120 vaccination as control. As the title of this manuscript is “Optimization of Infectious Bronchitis virus-like particle expression in Nicotiana benthamiana as potential poultry vaccines”. The study should include experiments to show the VLP as potential vaccines. The successful expression of S protein and HI experiment are not enough to demonstrate it is potential vaccines.

7. PLOS authors have the option to publish the peer review history of their article (what does this mean?). If published, this will include your full peer review and any attached files.

Reviewer #1: No

Reviewer #2: **Yes: **Liao Ying

---

## [Author Response · Author response to Decision Letter 1]

27 May 2023

The Academic Editor

PLOS ONE

27 May 2023

Dear Dr Haitham Mohamed Amer

Re. Requested further revisions to PONE-D-23-05238R1 -Optimization of Infectious

Bronchitis virus-like particle expression in Nicotiana benthamiana as potential poultry vaccines

by KS Sepotokele et al

Your letter to Ms. Sepotokele dated 23 May refers, in which you requested further changes to our

manuscript. Since the last revision, Reviewer 1 was satisfied with the revisions, but Reviewer 2

(Dr. Liao Ying) was not.

Whilst we are enormously grateful to Dr. Ying for the time expended on this review, we

unfortunately seem to have reached an impasse, and I explain below how and why this reviewer’s

three remaining objections lack merit. I have copied Dr. Ying’s comments in boldface, with our

responses below.

1. The author should re-perform the Western blot in Fig 1b and get better quality figure.

The current Western blot figure quality is not acceptable, as there is no clear S protein

band, even in the positive control (lane 3).

Ms. Sepotokele did repeat the Western blot several times, but she was unable to improve the quality.

The S protein band, although faint, is still clearly visible. The figure we present is the best we can

do at this time. It would be a shame to reject this novel work for this reason.

As we explained to the reviewer in our previous responses, included here for your benefit, the

reason for the faintness of the S-protein band in the Western blot is attributed to the fact that (a) the

plant extract containing the VLPs is only partially purified and plant proteins cross-react nonspecifically with chicken antiserum (chickens are fed plant matter, so this is to be expected) and (b)

that the only available IBV-positive antisera was collected from commercial chickens that were

immunized with heterologous IBV vaccines, not the QX-like variant our VLP is based on. Thus, repeating the Western blot over and over again is not going to increase the intensity of the S-protein

band whatsoever.

Finally, as we previously rebutted, peptide mass fingerprinting on this exact band confirmed that it

is indeed the S-protein.

Alternatively, if the acceptance of the manuscript comes down to this minor issue, we could remove

this Western Blot entirely, report it as “data not shown”; the peptide mass fingerprinting results are

already included as supplemental data. Please let us know if this would be acceptable?

2. S antibody measurement in the chicken serum after immunization is necessary.

We already explained this point in the prior revision, the hemagglutinin inhibition (HI) test

specifically measures S antibodies, so we did indeed measure the S antibodies in the chickens after

immunization; it is very strange that the reviewer won’t accept this fact.

3. The QX strain infection experiment after immunization is necessary to show the VLP

protection efficiency, by using the H120 vaccination as control. As the title of this manuscript

is “Optimization of Infectious Bronchitis virus-like particle expression in Nicotiana

benthamiana as potential poultry vaccines”. The study should include experiments to show

the VLP as potential vaccines. The successful expression of S protein and HI experiment are

not enough to demonstrate it is potential vaccines.

We completely disagree with Dr Ying’s view that successful expression of the S protein, and indeed

the proof we presented in the study that S-specific antibodies are raised in chickens immunized

with this VLP (as shown by HI), are insufficient to demonstrate the VLP’s potential as a vaccine.

It is very well known that the levels of coronavirus S-protein antibodies provide a predictive

correlate of protection in the host. With regard to the first sentence, the H120 vaccine is one of

many applied in the poultry industry globally, and is not a benchmark for establishing IBV vaccine

efficacy. Furthermore, the use of the word “Potential” in the title is deliberate.

We also, in our prior rebuttal, carefully explained that this study, which focused on optimising the

Gammacoronavirus VLP expression in planta (which, we point out again, is a world first) was

followed up by a vaccine-challenge study. For your confidential information, we attach a copy of

our submitted manuscript, to prove to you that these two manuscripts are intended to be

complimentary studies. There was too much information to describe all our processes and results

in a single paper, hence our decision to split it into an optimisation study (for a “potential”

vaccine), followed by the in vitro vaccine-challenge study with the optimised VLP.

To support our manuscript’s acceptance to PLOS ONE in its current form, we must point out that

two expert reviewers for PLOS ONE have already approved this manuscript. Ms Sepotokele is a

PhD candidate, and her thesis in which this work was presented has already been examined and

passed. I can confidentially disclose that the data in this manuscript was additionally reviewed

and approved by two external examiners (and an internal examiner). The two external examiners

were Prof. Janet Daly from Nottingham University, who is a leading world expert in veterinary

vaccine development (including plant VLPs), and Prof Inge Hitzeroth from the University of

Cape Town, who is a leading expert in plant-based vaccine development.

In conclusion, I am respectfully requesting that all of the abovementioned reasons and explanations

are taken into account, and that you exercise your discretion as the academic editor to make a final

decision on whether or not our manuscript will be accepted to PLOS ONE in its current form.

Yours sincerely,

Prof. Celia Abolnik

NRF-DSI SARChI- Poultry Health and Production

PhD Promotor

Email: celia.abolnik@up.ac.za

---

## [Decision Letter · Decision Letter 2]

19 Jun 2023

PONE-D-23-05238R2Optimization of Infectious Bronchitis virus-like particle expression in Nicotiana benthamiana as potential poultry vaccinesPLOS ONE

Dear Dr. Sepotokele,

Thank you for submitting your manuscript to PLOS ONE. After careful consideration, we feel that it has merit but does not fully meet PLOS ONE’s publication criteria as it currently stands. Therefore, we invite you to submit a revised version of the manuscript that addresses the points raised during the review process.

We look forward to receiving your revised manuscript.

Kind regards,

Haitham Mohamed Amer, PhD

Academic Editor

PLOS ONE

Journal Requirements:

Reviewers' comments:

Reviewer's Responses to Questions

**Comments to the Author**

1. If the authors have adequately addressed your comments raised in a previous round of review and you feel that this manuscript is now acceptable for publication, you may indicate that here to bypass the “Comments to the Author” section, enter your conflict of interest statement in the “Confidential to Editor” section, and submit your "Accept" recommendation.

Reviewer #1: All comments have been addressed

Reviewer #3: (No Response)

2. Is the manuscript technically sound, and do the data support the conclusions?

Reviewer #1: Yes

Reviewer #3: Yes

3. Has the statistical analysis been performed appropriately and rigorously? 

Reviewer #1: I Don't Know

Reviewer #3: Yes

4. Have the authors made all data underlying the findings in their manuscript fully available?

Reviewer #1: Yes

Reviewer #3: Yes

5. Is the manuscript presented in an intelligible fashion and written in standard English?

Reviewer #1: Yes

Reviewer #3: Yes

6. Review Comments to the Author

Reviewer #1: I was happy with the authors' modifications after the first round of reviews. I'm still ok with this manuscript being published.

Reviewer #3: This is a resubmission of a manuscript that has already been reviewed twice. From the correspondence associated with this latest version, it appears that although Reviewer 1 for second revision was happy with the changes made, Reviewer 2 (Dr. Liao Ying) was still not satisfied. Given that this manuscript has already been extensively reviewed by others, I will concentrate on the outstanding issues raised by Dr. Ying.

1. I disagree with the comment of Dr. Ying that there are no clear bands in the western blot in Fig. 1B. The bands corresponding to the full-length S protein are clearly visible in lanes 8-15, as are higher molecular weight forms which are commonly seen with multimeric proteins. The fact that the positive control does not give a clear band representing full-length S protein is really neither here nor there - it would only be an issue if no bands were seen in the test samples.

That said, I feel the authors do not do themselves any favours by not mentioning the absolutely obvious band at around 50kDa in lanes 4-7. This is also present in the positive control. This most probably represents the cleaved form of the S protein (S1 and/or S2) and such cleavage has been reported previously with plant expressed SARS-CoV-2 S protein (see Fig 2 oof reference [30]. The fact that this cleavage appears to be greatly reduced with the modified S protein (lanes 8-15) validates the authors' modification strategy and this should highlighted. At any rate the appearance of the 50kDa product must be discussed.

Two more minor points:

I do think that the S protein can be described as being comprised of two "subunits". This implies separately synthesised polypeptides which - the usual description is two "domains".

Fig.1 would greatly benefit from having a simple version of the constructs shown as an addition panel (C?). This would greatly assist the reader and avoid the necessity for constantly cross-referencing with the SI.

2. I have no problem with the immunological analysis as presented in the revised ms.

3. From the correspondence, it is clear the authors have additional information about the immunological properties of their VLPs which they wish to present in a separate MS. Though I appreciate their desire to present the work in two papers, I really thing the data would be better combined, especially as the current MS is so short (only 3 figures!). I think one MS would make far greater impact overall and I urge the authors to consider this.

7. PLOS authors have the option to publish the peer review history of their article (what does this mean?). If published, this will include your full peer review and any attached files.

Reviewer #1: No

Reviewer #3: **Yes: **George P. Lomonossoff

---

## [Author Response · Author response to Decision Letter 2]

26 Jun 2023

Response to Reviewers PLOS One

Reviewer #1: I was happy with the authors' modifications after the first round of reviews. I'm still ok with this manuscript being published.

RESPONSE: Thank you.

Reviewer #3: This is a resubmission of a manuscript that has already been reviewed twice. From the correspondence associated with this latest version, it appears that although Reviewer 1 for second revision was happy with the changes made, Reviewer 2 (Dr. Liao Ying) was still not satisfied. Given that this manuscript has already been extensively reviewed by others, I will concentrate on the outstanding issues raised by Dr. Ying.

1. I disagree with the comment of Dr. Ying that there are no clear bands in the western blot in Fig. 1B. The bands corresponding to the full-length S protein are clearly visible in lanes 8-15, as are higher molecular weight forms which are commonly seen with multimeric proteins. The fact that the positive control does not give a clear band representing full-length S protein is really neither here nor there - it would only be an issue if no bands were seen in the test samples.

That said, I feel the authors do not do themselves any favours by not mentioning the absolutely obvious band at around 50kDa in lanes 4-7. This is also present in the positive control. This most probably represents the cleaved form of the S protein (S1 and/or S2) and such cleavage has been reported previously with plant expressed SARS-CoV-2 S protein (see Fig 2 oof reference [30]. The fact that this cleavage appears to be greatly reduced with the modified S protein (lanes 8-15) validates the authors' modification strategy and this should highlighted. At any rate the appearance of the 50kDa product must be discussed.

RESPONSE: The bands present in lanes 4-7 at around 50 kDa (Western blot Fig. 1B), were thought to correlate to the highly conserved 45 kDa IBV Nucleocapsid protein, which would be present in the positive control because the positive control contains it. The reason this protein was only observed in lanes 4-7, was because those lanes contained the mIBV-S2P VLPs that had been produced by co-infiltrating with the IBV Membrane, Envelope, and Nucleocapsid proteins. It is not present in lanes 8-15 because constructs mIBV-S2P-IAV-H6TM/CT (8-11) and mIBV-S2P-NDV-FTM/CT (12-15) were not co-infiltrated with the IBV M, E, and N proteins, but co-infiltrated with accessory proteins corresponding to IAV and NDV instead.

This was discussed in lines 257-260. No further changes made. 

Two more minor points:

I do think that the S protein can be described as being comprised of two "subunits". This implies separately synthesised polypeptides which - the usual description is two "domains".

RESPONSE: “subunit/s” has been replaced with “domain/s” where necessary.

Fig.1 would greatly benefit from having a simple version of the constructs shown as an addition panel (C?). This would greatly assist the reader and avoid the necessity for constantly cross-referencing with the SI.

RESPONSE: Thank you for the suggestion. Fig 1. has now been amended to include an additional panel C, which contains the constructs. Fig S3 has accordingly been removed to avoid repetition.

2. I have no problem with the immunological analysis as presented in the revised ms.

RESPONSE: Thank you.

3. From the correspondence, it is clear the authors have additional information about the immunological properties of their VLPs which they wish to present in a separate MS. Though I appreciate their desire to present the work in two papers, I really thing the data would be better combined, especially as the current MS is so short (only 3 figures!). I think one MS would make far greater impact overall and I urge the authors to consider this.

RESPONSE: While we do understand this concern, the present manuscript covers the extensive work and technical details (multiple constructs and combinations of accessory proteins) that went into obtaining successful IBV VLP expression for the first time (which is novel), and we demonstrated the immunogenicity in chickens to prove that the S protein is properly folded and presented on the VLP. The second manuscript is a classical vaccine-efficacy study that is more suited to a veterinary journal, and we felt that with a single paper those reviewers might consider the in-depth description of how the VLP was developed of less interest to veterinarians and poultry scientists, and consequently request that the developmental data (that is of interest to plant scientists designing such VLPs) be removed. For this reason, we deemed it necessary to publish these aspects of the studies separately. 

The second manuscript that was submitted to “Poultry Science” has already been reviewed and we are busy preparing a revised version that incorporates those reviewers’ suggested changes, therefore we are unwilling to withdraw it at this stage in order to incorporate the data here (along with the other reason we describe above). We hope that Prof. Lomonossoff is satisfied with this explanation.

---

## [Decision Letter · Decision Letter 3]

10 Jul 2023

Optimization of Infectious Bronchitis virus-like particle expression in Nicotiana benthamiana as potential poultry vaccines

PONE-D-23-05238R3

Dear Dr. Sepotokele

We’re pleased to inform you that your manuscript has been judged scientifically suitable for publication and will be formally accepted for publication once it meets all outstanding technical requirements.

Kind regards,

Haitham Mohamed Amer, PhD

Academic Editor

PLOS ONE

---

## [Editor Report · Acceptance letter]

12 Jul 2023

PONE-D-23-05238R3 

Optimization of Infectious Bronchitis virus-like particle expression in *Nicotiana benthamiana* as potential poultry vaccines 

Dear Dr. Sepotokele:

I'm pleased to inform you that your manuscript has been deemed suitable for publication in PLOS ONE. Congratulations! Your manuscript is now with our production department. 

Kind regards, 

on behalf of

Dr. Haitham Mohamed Amer 

Academic Editor

PLOS ONE